# Loneliness Is Associated with Depressive Affect, But Not with Most Other Symptoms of Depression in Community-Dwelling Individuals: A Network Analysis

**DOI:** 10.3390/ijerph18052408

**Published:** 2021-03-01

**Authors:** Roland von Känel, Sonja Weilenmann, Tobias R. Spiller

**Affiliations:** Department of Consultation-Liaison Psychiatry and Psychosomatic Medicine, University Hospital Zurich, University of Zurich, CH-8006 Zurich, Switzerland; sonja.weilenmann@usz.ch (S.W.); tobias.spiller@usz.ch (T.R.S.)

**Keywords:** depression, depressive symptoms, loneliness, mental health, network analysis

## Abstract

There is a strong relationship between loneliness and depression, but depression is a heterogeneous disorder. We examined the profile of depressive symptoms most strongly related to loneliness. Study participants were 2007 community-dwelling individuals (median age 31 years, 70.4% women) who completed an online survey on loneliness (single-item question: “never”, “sometimes”, “often”), depressive symptoms (Patient Health Questionnaire-9) and demographics. The relationship between loneliness and depressive symptoms was evaluated with linear regression and network analyses. The prevalence of loneliness (sometimes or often) and of moderate depression was 47.1% and 24.0%, respectively. Loneliness explained 26% of the variance in the total depressive symptom score (*p* < 0.001), independent of covariates. This result was almost exclusively explained by the relationship with a single depression symptom (“feeling down, depressed, or hopeless”), irrespective of whether loneliness was treated as a nominal or continuous variable. The findings of our study suggest that the role of loneliness in depression should not only be investigated at the syndrome level, but also at the symptom level. Studies are warranted to test whether targeted treatment of depressive affect is particularly effective against loneliness.

## 1. Introduction

Loneliness is a subjective negative emotional state, resulting from a perceived deficit in the quantity or quality of a person’s network of social relationships [1], which has become a rampant public health concern. For example, the United Kingdom’s government has recently appointed a Minister of Loneliness to combat this “real and diagnosable scourge” [2]. Population-based studies in the Western hemisphere yield a prevalence between 10% and 50% of moderate-to-high loneliness [3,4,5]. An umbrella review of observational studies showed an association between loneliness and age (in a U-shaped way), female sex, low socioeconomic status and chronic medical conditions [6]. Recent research has confirmed the U-shaped or curvilinear association of loneliness with age, in that younger and much older individuals experience the most loneliness [7].

Loneliness is associated with numerous adverse health outcomes [8] with ample evidence of premature mortality, cardiovascular disease risk and depression. A meta-analysis showed an association of loneliness with an increased mortality risk of 26%, statistically controlling for several possible confounds, including pre-existent health conditions [9]. In 5397 men and women participating in the English Longitudinal Study of Ageing, loneliness was a predictor of incident coronary heart disease and stroke after a follow-up of 5 years, increasing the risk by 27%, independently of traditional cardiovascular risk factors [10]. Likewise, in 3003 women who participated in the First National Health and Nutrition Survey, loneliness was associated with a 76% increased risk of incident coronary heart disease after a follow-up of 19 years, controlling for a range of sociodemographics, cardiometabolic factors, health behaviors and depressive symptoms [11]. There is a strong relation between loneliness and depression, both cross-sectionally and longitudinally [6,12], even when controlling for demographic characteristics, marital status, psychosocial risk factors and social isolation [13], which needs to be distinguished from loneliness [9].

Despite a considerable overlap, loneliness and depressive symptoms are seen as distinct concepts [8,12]. However, depression is a heterogenous disorder with different clinical presentations, rendering the assumption that the diagnosis can be made merely on the number of depressive symptoms present an undue simplification [14]. Accordingly, there is an increasing number of research arguing that psychopathology needs to be investigated on an individual symptom level [15]. To our knowledge, the profile of depressive symptoms most strongly related to loneliness has not been investigated systematically. This information could have important clinical implications. For instance, depressive symptoms might contribute to seeking less social support and maladaptive social cognitions, which are starting points for effective interventions against loneliness [8]. In turn, although loneliness often precedes depression [6,8], depressive symptoms could also become a driving source of feeling lonely [16].

The aim of this study was to investigate the association of loneliness with (a) the depressive syndrome as a whole and (b) individual depressive symptoms in a sample of community-dwelling individuals from Switzerland. Thus, we aimed to investigate whether the known association of loneliness and depression mainly manifests on a syndrome level (with all symptoms contributing equally to this relationship) or on a symptom level, with a stronger relationship to some symptoms of depression than others. To investigate the latter, we applied network analysis as a novel approach to this field of research, which allows an estimate of all pairwise associations between the included variables with simultaneous adjustment for the effect of the other variables on a given association [17].

## 2. Materials and Methods

The data for this study were collected through an anonymous, nationwide online survey conducted in German, French, and Italian language from 9–14 May 2020, as part of a larger study investigating the mental health of healthcare workers and working controls not working in the healthcare sector during the SARS-CoV-2 pandemic in Switzerland. There was no requirement for ethical approval because the study did not fall within the scope of the Human Research Act (decision of the ethics committee of the canton Zurich; BASEC-Nr. Req-2020-00471). To be included in the larger study, participants had to be older than 18 years and younger than 70 years, which is the age of the latest official retirement in Switzerland. We recruited participants via mailing lists, institutional websites and personal contacts of the study team members. A total of 2077 participants did take part in the larger study. Due to the handling of the data for the network analysis, 3 (0.1%) participants who indicated their gender as “other” were excluded from the analysis. In addition, 67 participants (3.2%) with incomplete data on the variables of interest outlined below were excluded, resulting in a final sample size of 2007 individuals.

We assessed demographic factors (age, gender, living with children, language, profession), average number of sleep hours in the previous 7 days, loneliness and depressive symptoms. Demographic factors were collected using single-item questions using a binary option (yes/no), a list with multiple options (e.g., gender) or a continuous scale (e.g., hours of sleep). Loneliness was assessed with a direct single-item question asking, “How often did you feel lonely during the past 7 days” with response options “never”, “sometimes” or “often” [10]. Depressive symptoms in the last 7 days were assessed with the Patient Health Questionnaire-9 (PHQ-9; [18]). Each of the 9 symptoms is rated on a 4-point Likert scale ranging from 0 = “not at all” to 3 = “nearly every day”, yielding a sum score for depressive symptom severity between 0 and 27. Sum scores of 10 or higher indicate moderate (i.e., clinically relevant) depressive symptoms and have a sensitivity of 88% and a specificity of 88% for major depression [18].

To assess the relationship between loneliness and depression on a syndrome level, we calculated multivariable regression. In a first analysis, the PHQ-9 sum score was set as the dependent variable, and loneliness, age, gender, living with children and sleep hours per night as independent variables. We selected these potentially confounding variables as covariates based on the literature showing that the prevalence of depression is higher in women than men and decreases with age [19]. Moreover, shorter sleep duration has been associated with an increased risk of depression in community-dwelling middle-aged and elderly individuals [20]. Parents in a national sample reported higher levels of depression than nonparents when sociodemographic variables were held constant [21].

For the network analysis, we included age, gender, living with children (yes/no), the average hours of night sleep in the previous 7 days, loneliness (two dummy variables) and the nine symptoms of the PHQ-9. Prior to the network analysis itself, we assessed the overlap of the symptoms planned to be included into the network analysis using the standard settings of the goldbricker function of the networktools package (treating all variables as continuous). No exclusion of symptoms was suggested. Next, we estimated the network using a regularization technique based on LASSO (the least absolute shrinkage and selection operator) that sets very small edges to zero. This reduces the false positive rate of the analysis [22]. The network analysis and the stability and realizability analyses were carried out using the package bootnet [23]. In the resulting network, variables are represented by nodes, and edges between these variables represent statistical relationships adjusted for the effect of all other variables included in the network (in case of edges between two continuous variables, these relationships correspond to a partial correlation). The visualization of the network was conducted using the package qgraph.

Given that loneliness was assessed with a single-item question that does formally yield a score, we included loneliness as a nominal variable and therefore coded it with two dummy variables in all analyses. However, in a sensitivity analysis, we also performed the above outlined network analysis in which loneliness was treated as a continuous variable. All analyses were performed in the R-statistical environment using R version 3.6.1. 

## 3. Results

Table 1 shows the characteristics of the 2007 study participants that provided full information about the variables of interest.

With a median of 31 years (range 18–69 years), the sample was rather young, with twice as many women than men, and with one fourth of participants living together with children. The sizes of the language regions were fairly well represented. Whereas the median PHQ-9 sum score indicated mild depression on average, 482 (24.0%) participants scored 10 or higher, indicating moderate depression. About one in three participants (35.2%) felt sometimes lonely and one in eight (11.9%) felt often lonely. Regarding gender differences, women had a higher PHQ-9 total score and fewer women than men were German speaking in our sample. 

Table 2 presents the correlation matrix between the assessed variables.

Table 3 shows the results of the multivariable regression.

Compared to participants who felt never lonely, those who felt often lonely had a more than 8-point higher PHQ-9 sum score (*p* < 0.001); those who felt sometimes lonely had a more than 3-point higher PHQ-9 sum score (*p* < 0.001), adjusting for the other covariates in the model. Female gender and living with children were also significantly associated with more severe depressive symptoms, whereas age and sleep hours were not. The results were similar in the sensitivity analysis when loneliness was treated as a continuous variable (Appendix A). The result of the network analysis is shown in Figure 1 (and in Appendix A using the Fruchterman–Reingold algorithm for node placement).

There was a strong relationship between loneliness and item 2 of the PHQ-9, which is “feeling down, depressed or hopeless”. Except for a weak relationship with item 1 (“little interest or pleasure in doing things”) and 7 (“trouble concentrating on things”) of the PHQ-9, there emerged no significant and independent relationship between loneliness and any other depressive symptom. The result of the network analysis treating loneliness as a continuous variable was similar with additional weak relationships of loneliness with items 3 (“trouble falling or staying asleep, or sleeping too much”) and 6 (“feeling bad about yourself”; Appendix A). Further supplementary figures show the Bootstrap 95% confidence intervals of the edge weights of the network (Appendix A) and for the edge weights difference test between non-zero estimated edge-weights in the network (Appendix A) shown in Figure 1. The result of a relationship between loneliness and item 2 of the PHQ-9 was confirmed in a multivariable regression, controlling for the other eight PHQ-items (Appendix A). Moreover, a regression analysis with age as a linear and a quadratic term suggested a U-shaped association of age with loneliness (Appendix A).

## 4. Discussion

Our aim was to investigate the relationship between loneliness depression on a syndromal and an individual symptom level, applying the widely used PHQ-9 to allow a major depression diagnosis at a cut-off score of 10 with a sensitivity and specificity of 88% each [18]. In our sample of over 2000 people living in the community, 24% met this cut-off, indicating moderate depression, and 47% of participants felt lonely, consistent with other population surveys [3,4,5]. This prevalence suggested sufficient variation in the severity of loneliness and depression to allow a reliable analysis of their relationships.

We found that loneliness explained 26% of the variance in the total depressive symptom score that was based on all nine items of the PHQ-9, adjusting for age, gender, living with children and sleep hours per night. It made no difference whether we treated loneliness as a dummy or continuous variable. Expressed differently, participants who often felt lonely had an 8-point higher PHQ-9 total score compared with the nonlonely. This difference clearly exceeds the minimal clinical important difference of 5 points on the PHQ-9 [24]. To compare, although gender was also significantly associated with total depressive symptoms, as expected, women scored barely one point higher on the PHQ-9 total score than men. Likewise, only half a score point was significantly explained by whether a person lived with children or not. The latter observation could be understood in the context of the SARS-CoV-2 pandemic, as it is consistent with the results of a large study from England [25]. In that study, living with children was a risk factor for increased depressive symptoms, also measured with the PHQ-9, at the onset of forced isolation due to COVID-19 [25]. These authors argued that subsequent improvement in depressive symptoms may have been dependent on the level of information, suggesting that children were less affected by COVID-19. At the time we collected the data for the present study, there was still considerable controversy in the media in Switzerland regarding this issue. Another explanation could be the temporary closure of schools, which increased parental level of distress due to the forced care and schooling of children in the home environment. The relationship between less hours of sleep and greater severity of depressive symptoms was in the expected direction [20], but it did not reach statistical significance. On the whole, our finding of a robust and clinically highly relevant association between loneliness and depression, independent of covariates, is largely consistent with the literature [6,8,12].

Performing a network analysis, we found, as a novelty, that the association between loneliness and depression could be explained almost exclusively by the relationship with one single PHQ-9 item, reflecting depressive affect. This observation was robust, irrespective of whether loneliness was analyzed as a dummy or continuous measure, and independent of the relationship of loneliness with both demographic factors and the remaining PHQ-9 items. In agreement with a previous study [26], sleep duration was not associated with loneliness. Moreover, the result of a relationship between loneliness and item 2 of the PHQ-9 (“feeling down, depressed, or hopeless”) was also confirmed in a supplementary regression analysis. This observation suggests that the role of loneliness in depression should not only be investigated at the syndrome level but also at the symptom level. Moreover, studies on interactions between loneliness and depression should be careful to assume that depression is a latent factor, otherwise the association of loneliness with all nine depressive items should have turned out to be equally strong in our study. This was clearly not the case, in line with the growing recognition of depression as a heterogeneous disorder [14].

There may be several explanations for the prominent relationship of loneliness with the PHQ-9 item “feeling down, depressed or hopeless”, a core symptom of the depressive syndrome. Loneliness has been claimed to make people “sad” [13]. Although experiencing a similar number of uplifts throughout the day like nonlonely individuals, lonely people seem to experience less pleasure from these uplifts [27]. Finally, loneliness is a fundamental biological stressor, which might activate pro-inflammatory mechanisms [28], thereby inducing chronic sickness behavior of which depressive mood is a characteristic feature. Intervention studies could specifically target depressive mood in lonely people, for instance through modification of maladaptive social cognitions, with or without pharmacotherapy [8].

The large sample size, network analysis and use of established psychometric tools were strengths of our study, which also has notable limitations. The cross-sectional design does not allow conclusions to be drawn about the direction and causality of the relationship between loneliness and depressive symptoms. Data were not from a representative population sample and collected during a pandemic. The usual PHQ-9 format asks for the presence of symptoms over 14 days, the interval required to make a diagnosis of a depressive episode. We asked for symptoms in the previous seven days, so transient depressive reactions should be considered when interpreting our findings. Moreover, it has not been formally investigated whether the cut-offs and other psychometric properties of the original scale apply to our adapted version. As in previous population-based studies [10,11], we assessed loneliness with a one-item question, but multi-item questionnaires might be more reliable for measuring loneliness. Multi-item scales, such as the three-item Revised UCLA Loneliness Scale, which asks about the frequency of lack of companionship, feelings of being left out and of being isolated from others [29], would have provided more specific information about reasons for loneliness. For instance, as we collected our data during the SARS-CoV-2 pandemic, it may be that participants felt particularly lonely because of the lack of physical interaction with friends and family members. Our results must be interpreted with caution in relation to persons with major depression and older persons over 70 years of age. We controlled for living with children, which may, in the youngest study participants, refer to siblings rather than their children. Since the survey was to take only a few minutes, we lacked data on medical comorbidities and social isolation, covariates that are common in the loneliness literature.

## 5. Conclusions

In this community sample, the relationship between loneliness and depression was largely driven by depressive affect. To identify effective therapeutic interventions, longitudinal studies in clinical samples are warranted to better understand the temporal relationship and generalizability of this observation.

## Figures and Tables

**Figure 1 ijerph-18-02408-f001:**
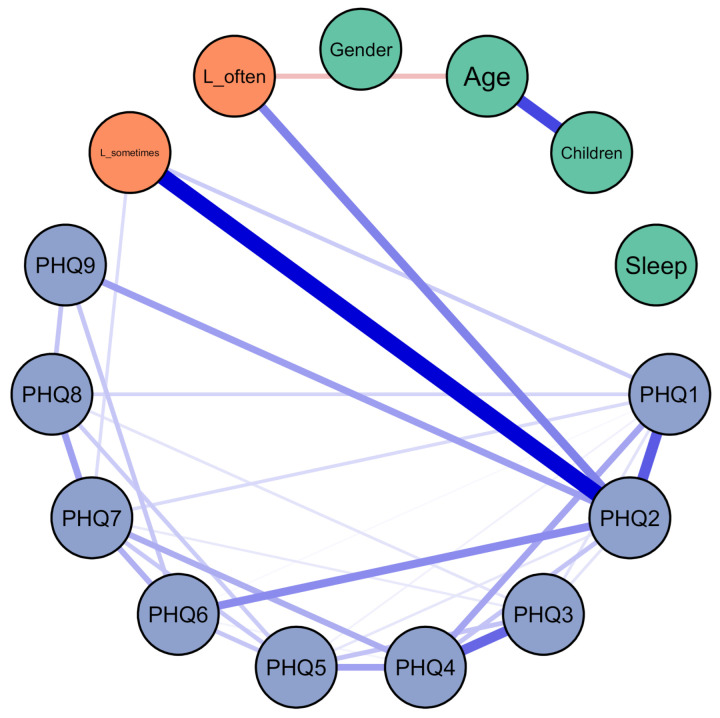
Relationships between PHQ-9 items, demographics and loneliness. Green: demographics; orange: loneliness; blue: PHQ; the edge between L_sometimes and L_often was excluded. L_sometimes = dummy variable encoding loneliness “sometimes” as 1, “never” or “often” as 0; L_often = dummy variable encoding loneliness “often” as 1, “never” or “sometimes” as 0; children = living with children in the same household; sleep = average hours of sleep per night in the previous 7 days; Patient Health Questionnaire (PHQ)-1 = little interest or pleasure in doing things; PHQ-2 = feeling down, depressed, or hopeless; PHQ-3 = trouble falling or staying asleep, or sleeping too much; PHQ-4 = feeling tired or having little energy; PHQ-5 = poor appetite or overeating; PHQ-6 = feeling bad about yourself—or that you are a failure or have let yourself or your family down; PHQ-7 = trouble concentrating on things, such as reading the newspaper or watching television; PHQ-8 = moving or speaking so slowly that other people could have noticed. Or the opposite—being so fidgety or restless that you have been moving around a lot more than usual; PHQ-9 = thoughts that you would be better off dead, or of hurting yourself.

**Table 1 ijerph-18-02408-t001:** Characteristics of the 2007 study participants.

Variable	Overall	Females (n = 1413)	Males (n = 594)	*p*-Value
Age, years (median IQR)	31 (24–41)	31 (24–42)	30 (25–38)	0.202 ^§^
Female gender, n (%)	1413 (70.4)			
Spoken language				<0.001
German, n (%)	901 (44.9)	667 (47.2)	234 (39.4)	
French, n (%)	955 (47.6)	657 (46.5)	298 (50.2)	
Italian, n (%)	151 (7.5)	89 (6.3)	62 (10.4)	
Living with Children, n (%)	559 (25.3)	398 (28.2)	161 (27.2)	0.628
Average number of sleep hours in the previous 7 days (median, IQR)	7 (6–8)	7 (6–8)	7 (6–8)	0.853 ^§^
Patient Health Questionnaire-9, total score (median, IQR)	5 (2–9)	6 (3–10)	5 (2–8)	<0.001 ^§^
Loneliness				0.280
Never, n (%)	1062 (52.9)	734 (51.9)	328 (55.2)	
Sometimes, n (%)	706 (35.2)	502 (35.5)	204 (34.3)	
Often, n (%)	239 (11.9)	177 (12.5)	62 (10.4)	

IQR, inter-quartile range. Between-group comparisons using chi-square for categorical variables and Mann–Whitney U test (§) for continuous variables.

**Table 2 ijerph-18-02408-t002:** Zero order correlation matrix.

Variable	1	2	3	4	5	6	7
1. Gender (f)	-						
2. Age	0.03	-					
3. Children	0.01	0.32 **	-				
4. Sleep (h)	0.00	−0.23 **	−0.05 *	-			
5. PHQ9	0.10 **	−0.07 **	0.00	−0.20 **	-		
6. L_some ^a^	0.03	−0.07 **	−0.06 **	−0.09 **	0.38 **	-	
7. L_often ^b^	0.01	−0.13 **	−0.06 **	−0.06 **	0.21 **	−0.27 **	-

PHQ9-Patient Health Questionnaire-9. Spearman coefficients are shown. f, female; h, hours; L, loneliness; ^a^ L_some = dummy variable encoding loneliness “sometimes” as 1, “never” or “often” as 0; ^b^ L_often = dummy variable encoding loneliness “often” as 1, “never” or “sometimes” as 0. * *p*-value < 0.050, ** *p*-value < 0.010.

**Table 3 ijerph-18-02408-t003:** Associations with overall Patient Health Questionnaire-9 score.

Effect	Estimate		SE	95% CI	*p*
			LL	UL	
Intercept	4.02	0.392	3.25	4.79	<0.001
Gender (f)	0.81	0.216	0.38	1.23	<0.001
Age	−0.01	0.009	−0.02	0.01	0.546
Children	0.47	0.229	0.02	0.92	0.039
Sleep (h)	−0.03	0.017	−0.06	0.00	0.093
L_some ^a^	3.09	0.217	2.67	3.52	<0.001
L_often ^b^	8.31	0.318	7.69	8.93	<0.001

Values are unstandardized beta coefficients with standard error (SE) and 95% confidence interval (CI); f, female; h, hours; L, loneliness; ^a^ L_some = dummy variable encoding loneliness “sometimes” as 1, “never” or “often” as 0; ^b^ L_often = dummy variable encoding loneliness “often ” as 1, “never” or “sometimes” as 0; F-statistic: 129.1 on 6 and 2000 degrees of freedom, *p*-value: < 0.001; adjusted R2 overall = 0.277; adjusted R2 change due to loneliness = 0.265.

## Data Availability

The anonymized data that support the findings of this study are available from the corresponding author on reasonable request.

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
