# Peer review of "Loneliness Is Associated with Depressive Affect, But Not with Most Other Symptoms of Depression in Community-Dwelling Individuals: A Network Analysis"

_ijerph, 2021, doi:10.3390/ijerph18052408_

Round 1
Reviewer 1 Report
This manuscript has a good potential. Nevertheless, a few things need to be more clarified.
- Abstract should be written according to IJERPH propositions (also the last chapter of it should consider Conclusions instead of Discussion)
- The whole Methods section should be broaden a little bit and better explained (how the authors choose their sample from "part of a larger study"; measures ; "analyses were carried out in R version 3.6.1." - some more details about this software)
- Table 2. presenting multivariable linear regression analysis was not cited in further text.
- The whole paragraph under Figure 1. "The edge between L_sometimes and L_often was excluded. L_sometimes = Dummy variable encoding loneliness “sometimes” as 1, “never” or “often” as 0 ... PHQ-9 = Thoughts that you would be better off dead, or of hurting yourself." should be marked as legend.
Author Response
Comment 1
This manuscript has a good potential. Nevertheless, a few things need to be more clarified.
Response
Thank you for the favorable appraisal of our paper
Comment 2
Abstract should be written according to IJERPH propositions (also the last chapter of it should consider Conclusions instead of Discussion)
Response
We apologize for this oversight. We removed the terms “Background”, “Methods”, Results” and “Discussion” from the abstract and formulated the second last sentence more as a conclusion.
Comment 3
The whole Methods section should be broaden a little bit and better explained (how the authors choose their sample from "part of a larger study"; measures ; "analyses were carried out in R version 3.6.1." - some more details about this software)
Response
We agree with the reviewer that the methods section could have been more detailed. We broadened it by adding the following information:
On Page 2:
A total of 2077 participants did take part in the larger study. Due to the handling of the data for the network analysis 3 (0.1%) participants who indicated their gender as “other” were excluded from the analysis. In addition, 67 participants (3.2%) with incomplete data on the variables of interest outlined below were excluded, resulting in a final sample size of 2007 individuals.
Demographic factors were collected using single item questions using a binary option (yes/no), a list with multiple options (e.g., gender) or a continuous scale (e.g., hours of sleep).
On page 3:
We selected these potentially confounding variables as covariates based on the literature showing that the prevalence of depression is higher in women than men and decreases with age [19]. Moreover, shorter sleep duration has been associated with an increased risk of depression in community-dwelling middle-aged and elderly individuals [20]. Parents in a national sample reported higher levels of depression than nonparents when sociodemographic variables are held constant [21].
For the network analysis, we included age, gender, living with children (yes/no), the average hours of night sleep in the previous 7 days, loneliness (two dummy variables) and the nine symptoms of the PHQ-9. Prior to the network analysis itself, we assessed the overlap of the symptoms planned to be included into the network analysis using the standard settings of the goldbricker function of the networktools package (treating all variables as continuous). No exclusion of symptoms was suggested. Next, we estimated the network using a regularization technique based on LASSO (the least absolute shrinkage and selection operator) that sets very small edges to zero. This reduces the false positive rate of the analysis [22]. The network analysis and the stability and realizability analyses were carried out using the package bootnet [23]. In the resulting network, variables are represented by nodes, and edges between these variables represent statistical relationships adjusted for the effect of all other variables included in the network (in case of edges between two continuous variables, these relationships correspond to a partial correlation). The visualization of the network was conducted using the package qgraph.
Given that loneliness was assessed with a single-item question that does formally yield a score, we included loneliness as a nominal variable and therefore coded it with two dummy variables in all analyses. However, in a sensitivity analysis, we also performed the above outlined network analysis in which loneliness was treated as a continuous variable. All analyses were performed in the R-statistical environment using R version 3.6.1.
Comment 4
Table 2. presenting multivariable linear regression analysis was not cited in further text.
Response
We cite Table 2 (now Table 3) in the Result section (Page 4) as follows: “Table 3 shows the results of the multivariable regression analysis.”
Comment 5
The whole paragraph under Figure 1. "The edge between L_sometimes and L_often was excluded. L_sometimes = Dummy variable encoding loneliness “sometimes” as 1, “never” or “often” as 0 ... PHQ-9 = Thoughts that you would be better off dead, or of hurting yourself." should be marked as legend.
Response
We assume that this issue occurred due to the automated formatting for the text layout of the journal. We have now combined this text as one legend to Figure 1, as requested by the Reviwer.

Reviewer 2 Report
This was an interesting paper to read. I would want to see the correlation matrix presented in the results section. That should be Table 2.
The authors are not clear in their participants section as to why they excluded individuals older than 70. It is also unclear why the curvilinear relationship between age and loneliness was not double-checked as that relationship is robust and it would help to verify that the results are "behaving" as previous researchers have reported, such as:
MacDonald et al. Soc. Sci. 2020, 9, 51; doi:10.3390/socsci9040051
Also, it is unclear why "living with children" was used as a variable if the authors tested individuals as young as 18 years old. An 18 year old individual is more likely to be living with their parents and not with children (unless those children are siblings). Please add this fact in the limitations section as it is a bit short sighted.
I'm glad that the single item loneliness item was highlighted as a limitation. I would have liked to have seen a bit more speculation with respect to multi-item loneliness measures. For example, as this data was collected during the pandemic, were individuals more lonely because of a lack of physical interaction with friends and family members?
Author Response
Comment 1
This was an interesting paper to read. I would want to see the correlation matrix presented in the results section. That should be Table 2.
Response
We agree with the Reviewer that a correlation matrix would be informative in terms of unadjusted associations between the variables of interest. We added a novel Table 2 with zero order correlation coefficients to the Results (page 4).
Comment 2
The authors are not clear in their participants section as to why they excluded individuals older than 70.
Response
As the larger study investigated the mental health of health-care workers and controls not working in the health care sector during the SARS-CoV-2 pandemic, we only included participants who were potentially still in the workforce. We added the following information to the Materials and Methods section, page 2, 1st paragraph:
To be included in the study, participants had to be older than 18 years and younger than 70 years, which is the age of the latest official retirement in Switzerland.
Comment 3
It is also unclear why the curvilinear relationship between age and loneliness was not double-checked as that relationship is robust and it would help to verify that the results are "behaving" as previous researchers have reported, such as: MacDonald et al. Soc. Sci. 2020, 9, 51; doi:10.3390/socsci9040051
Response
We thank the Reviewer for bringing up this valuable point. In the Introduction, we had already mentioned that there is a U-shaped association between loneliness and age in observational studies, citing the recently published umbrella review by Solmi et al (ref 6). This umbrella review included 14 reviews published up to 2018; therefore, the mentioned study by MacDonald et al 2020 was not part of it.
We added the following information to the Introduction (page 1), now also citing the Mac Donald et al study (ref 7):
Recent research has confirmed the U-shaped or curvilinear association of loneliness with age, in that younger and much older individuals experience the most loneliness [7].
We additionally investigated the U-shaped association of age with loneliness using a regression analysis with age as a linear and a quadratic term and provide the results in Table S3. Both, age and age as a quadratic term were significant predictors of loneliness. However, recent methodological studies suggest that regression analysis with quadratic terms is not sufficient to assess the presence of a U-shaped relationship between two variables, but suggest other tests (see Simonsohn, U. (2018). Two lines: A valid alternative to the invalid testing of U-shaped relationships with quadratic regressions. Advances in Methods and Practices in Psychological Science, 1(4), 538-555.). However, these tests are not available for multivariable regression. Thus, we decided to only report the results of the “classic” way of analyzing U-shaped relationships in the supplementary materials (Table S3).
In the revised manuscript, we added the following text to the Results (bottom of page 5):
Moreover, a regression analysis with age as a linear and a quadratic term suggested a U-shaped association of age with loneliness (Table S3).
Comment 4
Also, it is unclear why "living with children" was used as a variable if the authors tested individuals as young as 18 years old. An 18 year old individual is more likely to be living with their parents and not with children (unless those children are siblings). Please add this fact in the limitations section as it is a bit short sighted.
Response
We thank the Reviewer for bringing up this limitation. For the survey, we decided to ask for “living with children” as an umbrella term. We acknowledge that this simplification comes with costs of imprecision. We added the following sentence to the limitations section (p. 7):
We controlled for living with children, which may, in the youngest study participants, refer to siblings rather than their children.
In the Materials and Methods section, we added the information about why we used “living with children” as a control variable (page 3, 2nd paragraph):
We selected these potentially confounding variables as covariates based on the literature showing that the prevalence of depression is higher in women than men and decreases with age [19]. Moreover, shorter sleep duration has been associated with an increased risk of depression in community-dwelling middle-aged and elderly individuals [20]. Parents in a national sample reported higher levels of depression than nonparents when sociodemographic variables are held constant [21].
Comment 5
I'm glad that the single item loneliness item was highlighted as a limitation. I would have liked to have seen a bit more speculation with respect to multi-item loneliness measures. For example, as this data was collected during the pandemic, were individuals more lonely because of a lack of physical interaction with friends and family members?
Response
This is a valuable point. We added the following sentences to the limitations section (page 7, 2nd paragraph):
Multi-item scales, such as the 3-item Revised UCLA Loneliness Scale, which asks about the frequency of lack of companionship, feelings of being left out and of being isolated from others [29], would have provided more specific information about reasons for loneliness. For instance, as we collected our data during the SARS-CoV-2 pandemic, it may be that participants felt particularly lonely because of the lack of physical interaction with friends and family members.

Reviewer 3 Report
The manuscript describes an investigation of the association between loneliness and depression in a community sample. Researchers employed network analysis and linear regression to assess relationships in the sample of n=2,700 individuals. The study showed that loneliness is more prevalent in moderately depressed patients and explained 26% of the variance in the depression score. Researchers suggested that loneliness should be investigated at both symptom and syndrome levels.
Abstract:
- In general, this section is exceptionally well written and informative. I am not sure if information about regression covariates is needed.
Introduction:
- “association between loneliness and age (in a U-shaped way)” - U-shaped is not the clearest description, statement, whether age was or was not a risk factor, will make the message more lucid.
- “depression is a heterogeneous construct with different clinical presentations” - does it refer to the heterogeneity of major depressive disorder (MDD) or depression as a set of depressive symptoms across diagnoses? I am not sure if “construct” is the optimal word here, suggesting depression is artificial.
Methods:
- “multivariable linear regression analyses” - this is long, multivariate regression is sufficient.
- “ age, gender, living with children and sleep hours per night as independent variables” - were covariates selected based on expert knowledge and/or previous studies or covariate selection was conducted?
Results:
- Depression is observed significantly more often in females than in males. I wonder if loneliness, sleep, and PHQ scores differed between the sexes. Table1 summarizes the whole sample, but I am curious about males and females.
- Table 2 provides the results of the association study, not a prediction.
- For a network of small size different layouts, including sphere or fruchterman.reingold, provide desirable visualization (in igraph and ggally packages of R).
Discussion:
- Female sex is a known risk factor for depression, but I wonder if other groups found an association between living with children (and n of children) as a risk factor. If not, I wonder what can be the possible explanation (covid effect)?
- “Performing a network analysis, we found, as a novelty, that the association between loneliness and depression could be explained almost exclusively by the relationship with one single PHQ-9 item, reflecting depressive affect. “ - true, but for making such a strong statement I would fit multivariate regression using this item as a dependent variable. On the network level association seems strong, but I wonder what is the level (p-value), strength, and magnitude (regression coefficient).
- “ Moreover, studies on interactions between loneliness and depression should not assume that depression is a latent factor, otherwise the association of loneliness with all nine depressive items should have turned out to be equally strong in our study.” - without a series of multiple regressions, this is a very strong statement. I strongly suggest fitting a series of follow-up regressions to assess associations between separate PHQ-9 items and loneliness.
- “The large sample size, sophisticated statistics, and use of established psychometric tools were strengths of our study “ - the sample size is indeed large, but multivariate linear regression is a standard statistical tool.
- “ Our results cannot be generalized to populations with a clinical diagnosis of major depression and elderly people” - this is again a very strong statement. Are there any studies suggesting the association between loneliness and depressive symptoms is different in MDD and/or elderly populations? If so, the reference would be a plus. If not, I will rather write something like “Results need to be interpreted with caution in regards to MDD and/or elderly population”. Also, participants were below 70 years old making elderly people already part of a study (for example late-life depression is diagnosed in years 60+)
Author Response
Comment 1
The manuscript describes an investigation of the association between loneliness and depression in a community sample. Researchers employed network analysis and linear regression to assess relationships in the sample of n=2,700 individuals. The study showed that loneliness is more prevalent in moderately depressed patients and explained 26% of the variance in the depression score. Researchers suggested that loneliness should be investigated at both symptom and syndrome levels.
Response
Thank you for summarizing the main content of our manuscript so accurately.
Comment 2
Abstract: In general, this section is exceptionally well written and informative. I am not sure if information about regression covariates is needed.
Response
Thank you for this favorable comment. We deleted the information on individual covariates in the abstract. However, to clarify that the association was independent of covariates, we rephrased this particular sentence in the Abstract as follows:
Loneliness explained 26% of the variance in the total depressive symptom score (p<.001), independent of covariates.
Comment 3
Introduction: “association between loneliness and age (in a U-shaped way)” - U-shaped is not the clearest description, statement, whether age was or was not a risk factor, will make the message more lucid.
Response
Please see also our response to comment 3 from Reviewer 2 on the epidemiological observation of U-shaped association between age and loneliness. We clarified the description “U-shaped association” in terms of age as a risk factor with the following sentence in the Introduction (page 1):
Recent research has confirmed the U-shaped or curvilinear association of loneliness with age, in that younger and much older individuals experience the most loneliness [7].
Comment 4
Introduction: “depression is a heterogeneous construct with different clinical presentations” - does it refer to the heterogeneity of major depressive disorder (MDD) or depression as a set of depressive symptoms across diagnoses? I am not sure if “construct” is the optimal word here, suggesting depression is artificial.
Response
We agree with the Reviewer. We replaced “construct” by “disorder” throughout the revised manuscript.
Comment 5
Methods: “multivariable linear regression analyses” - this is long, multivariate regression is sufficient.
Response
Thank you for this comment. We changed “multivariable linear regression analyses” to “multivariable regression” throughout the revised manuscript. Please note that we favor the term “multivariable” over “multivariate” as all regression models considered just one dependent variable. To our statistical knowledge, the term “multivariate analysis” is usually recommended when referring to statistical models that have 2 or more dependent or outcome variables (e.g.: doi:10.2105/AJPH.2012.300897).
Comment 6
Methods: “ age, gender, living with children and sleep hours per night as independent variables” - were covariates selected based on expert knowledge and/or previous studies or covariate selection was conducted?
Response
We agree that this information was lacking. We added the rational for the selection of these covariates in our statistical models as follows (Materials and Methods section, page 7, 2nd paragraph):
We selected these potentially confounding variables as covariates based on the literature showing that the prevalence of depression is higher in women than men and decreases with age [19]. Moreover, shorter sleep duration has been associated with an increased risk of depression in community-dwelling middle-aged and elderly individuals [20]. Parents in a national sample reported higher levels of depression than nonparents when sociodemographic variables are held constant [21].
Comment 7
Results: Depression is observed significantly more often in females than in males. I wonder if loneliness, sleep, and PHQ scores differed between the sexes. Table1 summarizes the whole sample, but I am curious about males and females.
Response
We agree with the Reviewer that there are expected differences between males and females in several sample characteristics shown in Table 1 (page 3). In addition to the characteristics of the entire sample of 2007 study participants, we added two columns showing these characteristics by gender (males: n= 594; females: n=1413) along with p-values for the group comparison.
We added the following information to the Results summarizing gender differences in participant characteristics (page 4):
Regarding gender differences, women had a higher PHQ-9 total score and fewer women than men were German speaking in our sample.
Comment 8
Results: Table 2 provides the results of the association study, not a prediction.
Response
Thank you for this specification. We changed the title of Table 2 – now Table 3 on page 4 - from “Prediction of overall Patient Health Questionnaire-9 score.” to “Associations with overall Patient Health Questionnaire-9 score”.
Comment 9
Results: For a network of small size different layouts, including sphere or fruchterman.reingold, provide desirable visualization (in igraph and ggally packages of R).
Response
We additionally provided the Fruchterman-Reingold visualization in Suppelementary Figure 1. However, we think that the current circular visualization is more readable for the general audience not familiar with network analysis.
In the Results section, bottom of page 4, we added the following sentence:
The result of the network analysis is shown in Figure 1 (and in Figure S1 using the Fruchterman-Reingold algorithm for node placement).
Comment 10
Discussion: Female sex is a known risk factor for depression, but I wonder if other groups found an association between living with children (and n of children) as a risk factor. If not, I wonder what can be the possible explanation (covid effect)?
Response
Thank you for this interesting comment. It could indeed be that our finding of a positive association between living with children and increased levels of depressive symptoms can be understood in the context of the pandemic. Our finding is actually consistent with a large study with 70,000 adults from England. Accordingly, we added the following sentences to the Discussion, page 6, 2nd paragraph:
The latter observation could be understood in the context of the SARS-CoV-2 pandemic, as it is consistent with the results of a large study from England [25]. In that study, living with children was a risk factor for increased depressive symptoms, also measured with the PHQ-9, at the onset of forced isolation due to COVID-19 [25]. These authors argued that subsequent improvement in depressive symptoms may have been dependent on level of information, suggesting that children were less affected by COVID-19. At the time we collected the data for the present study, there was still considerable controversy in the media in Switzerland regarding this issue. Another explanation could be the temporary closure of schools, which increased parental level of distress due to the forced care and schooling of children in the home environment.
Comment 11
Discussion: “Performing a network analysis, we found, as a novelty, that the association between loneliness and depression could be explained almost exclusively by the relationship with one single PHQ-9 item, reflecting depressive affect. “ - true, but for making such a strong statement I would fit multivariate regression using this item as a dependent variable. On the network level association seems strong, but I wonder what is the level (p-value), strength, and magnitude (regression coefficient).
Response
Conceptually, the network analysis can be understood like a series of regression analyses, in which each node is once the outcome and all other variables in the network are the predictors. However, there are two important extensions. First, some nodes are binary. Thus, in these instances the regression model would be a logistic one. Given that in the network analysis, all parameters are standardized, and the model is estimated as a whole, it formally is a multi-level model (and not a series of linear regressions). Second, to reduce the number of false positive results, the estimation of the network uses a penalization (LASSO). This has not been made explicit in the manuscript before and this is why we have revised the Methods section accordingly. However, conducting a series of multivariate or multivariable regressions would likely lead to parameters that could not be standardized easily across all models. Moreover, one would certainly need to correct for the FDR, which, in the end, is then a very similar procedure as conducting a network analysis.
Nevertheless, we conducted a multivariable regression analysis with PHQ-2 as the outcome and all other variables (including the other eight PHQ items) as predictors. Loneliness was a significant predictor of PHQ-2. We included the results in the supplementary materials (Table S2)
In the Materials and Methods section, page 3, 3rd paragraph, we added the following specification regarding the network analysis:
For the network analysis, we included age, gender, living with children (yes/no), the average hours of night sleep in the previous 7 days, loneliness (two dummy variables) and the nine symptoms of the PHQ-9. Prior to the network analysis itself, we assessed the overlap of the symptoms planned to be included into the network analysis using the standard settings of the goldbricker function of the networktools package (treating all variables as continuous). No exclusion of symptoms was suggested. Next, we estimated the network using a regularization technique based on LASSO (the least absolute shrinkage and selection operator) that sets very small edges to zero. This reduces the false positive rate of the analysis [22]. The network analysis and the stability and realizability analyses were carried out using the package bootnet [23]. In the resulting network, variables are represented by nodes, and edges between these variables represent statistical relationships adjusted for the effect of all other variables included in the network (in case of edges between two continuous variables, these relationships correspond to a partial correlation). The visualization of the network was conducted using the package qgraph.
In the Results section, bottom of page 5, we added the following information:
The result of a relationship between loneliness and item 2 of the PHQ-9 was confirmed in a multivariable regression, controlling for the other eight PHQ-items (Table S2).
Finally, in the Discussion section, page 6, 3rd paragraph, we added:
Moreover, the result of a relationship between loneliness and item 2 of the PHQ-9 (“feeling down, depressed, or hopeless”) was also confirmed in a supplementary regression analysis.
Comment 12
Discussion: “ Moreover, studies on interactions between loneliness and depression should not assume that depression is a latent factor, otherwise the association of loneliness with all nine depressive items should have turned out to be equally strong in our study.” - without a series of multiple regressions, this is a very strong statement. I strongly suggest fitting a series of follow-up regressions to assess associations between separate PHQ-9 items and loneliness.
Response
We agree with the reviewer that this is a strong statement and we toned it down accordingly in the Discussion, page 6, 3rd paragraph:
Moreover, studies on interactions between loneliness and depression should be careful to assume that depression is a latent factor, otherwise the association of loneliness with all nine depressive items should have turned out to be equally strong in our study.
However, given the conceptual similarity of network analysis with the requested series of regression analyses (see our response to the previous comment 11), we decided not to include them. Still, if the reviewer insists of including them, we would certainly be willing to do so.
Comment 13
Discussion: “The large sample size, sophisticated statistics, and use of established psychometric tools were strengths of our study “ - the sample size is indeed large, but multivariate linear regression is a standard statistical tool.
Response
The Reviewer is of course correct regarding our use of regression analysis. We apologize for being imprecise. We actually meant to relate this statement to the applied network analysis. We are not aware of any previous study applying network analysis with the aim to better understand the relationship between loneliness and depression in a large population-based study. We edited this sentence accordingly in the Discussion, page 7, 2nd paragraph:
The large sample size, network analysis and use of established psychometric tools were strengths of our study, which also has notable limitations.
Comment 14
Discussion: “ Our results cannot be generalized to populations with a clinical diagnosis of major depression and elderly people” - this is again a very strong statement. Are there any studies suggesting the association between loneliness and depressive symptoms is different in MDD and/or elderly populations? If so, the reference would be a plus. If not, I will rather write something like “Results need to be interpreted with caution in regards to MDD and/or elderly population”. Also, participants were below 70 years old making elderly people already part of a study (for example late-life depression is diagnosed in years 60+)
Response
The reviewer is correct. Based on the literature, no clear conclusion can be drawn as to whether the association between loneliness and depressive symptoms is different in MDD and/or older populations. Therefore, we rephrased this statement as suggested by the Reviewer; Discussion, page 7, 2nd paragraph:
Our results must be interpreted with caution in relation to persons with major depression and older persons over 70 years of age.

Round 2
Reviewer 1 Report
Dear authors, I suppose it is Ok now to be published.
Congrats!